# Elastic dosage compensation by X-chromosome upregulation

Antonio Lentini [1], Huaitao Cheng [2], J. C. Noble[1], Natali Papanicolaou[1], Christos Coucoravas[1], Nathanael Andrews [2], Qiaolin Deng [3], Martin Enge [2] & Björn Reinius [1✉]

X-chromosome inactivation and X-upregulation are the fundamental modes of chromosome-wide gene regulation that collectively achieve dosage compensation in mammals, but the regulatory link between the two remains elusive and the X-upregulation dynamics are unknown. Here, we use allele-resolved single-cell RNA-seq combined with chromatin accessibility profiling and finely dissect their separate effects on RNA levels during mouse development. Surprisingly, we uncover that X-upregulation elastically tunes expression dosage in a sex- and lineage-specific manner, and moreover along varying degrees of X-inactivation progression. Male blastomeres achieve X-upregulation upon zygotic genome activation while females experience two distinct waves of upregulation, upon imprinted and random X-inactivation; and ablation of *Xist* impedes female X-upregulation. Female cells carrying two active X chromosomes lack upregulation, yet their collective RNA output exceeds that of a single hyperactive allele. Importantly, this conflicts the conventional dosage compensation model in which naïve female cells are initially subject to biallelic X-upregulation followed by X-inactivation of one allele to correct the X dosage. Together, our study provides key insights to the chain of events of dosage compensation, explaining how transcript copy numbers can remain remarkably stable across developmental windows wherein severe dose imbalance would otherwise be experienced by the cell.

[1] Department of Medical Biochemistry and Biophysics, Karolinska Institutet, Stockholm, Sweden. [2] Department of Oncology and Pathology, Karolinska Institutet, Stockholm, Sweden. [3] Department of Physiology and Pharmacology, Karolinska Institutet, Stockholm, Sweden. ✉email: bjorn.reinius@ki.se

n most mammals, the X chromosome is present as two copies in females but only one in males, and two X-chromosome-wide mechanisms ensure balanced expression dosage in the cell[1]. X-chromosome upregulation (XCU) evolved to resolve X-to-autosomal imbalances in XY males by hyperactivation of the single X-chromosome whereas X-chromosome inactivation (XCI) followed as a mechanism to silence one hyperactive allele in XX females, equalizing expression between the sexes[1]. This stands as the prevailing evolutionary hypothesis of mammalian dosage compensation[2–4]. In addition, XCU followed by XCI is also believed to be the developmental sequence of events. Specifically, the conventional model of mammalian dosage compensation assumes an initial state of biallelically upregulated X chromosomes in female embryonic cells upon which XCI subsequently corrects the expression dosage by silencing one X allele. However, despite being central for the mechanistic understanding of how cells achieve dosage compensation, the developmental dynamics of XCU have so far not been thoroughly characterized in a mammal. Previous studies have approximated XCU primarily by the relative measurement between nonallelic total expression levels of X and autosomes in steady-state XCU[5–10]. Such comparisons are however not only indirect since autosomes and sex chromosomes differ in multiple aspects such as gene content but also confounded as opposite dosage effects of XCU and XCI on each allele would be masked at a nonallelic level. It is therefore unsurprising that reports on the establishment, magnitude, maintenance, and potential reversal of XCU have been conflicting[5,11,12]. Disentangling the isolated dosage effects of XCU and XCI requires quantitative gene expression measurements at a cellular and allelic resolution only recently enabled by allele-resolved single-cell RNA-seq (scRNA-seq), and computationally accounting for stochastic processes affecting allelic measurements at the cellular level[13], such as effects of transcriptional bursting and random XCI (rXCI) progressing heterogeneously in cells.

Here, we reveal the dynamics of XCU in mouse at the cellular and allelic resolution throughout mouse embryonic stem cell (mESC) priming in vitro as well as pre- and peri-implantation development in vivo. By combining allele-resolved scRNA-seq and scATAC-seq (single-cell assay for transposase-accessible chromatin using sequencing) in the same cells during XCU establishment, we directly contrast epigenetic features of the hyperactive and non-hyperactive state for the first time. Remarkably, our data show that XCU is neither a constant state of hyper transcription of the X chromosome, nor established in a single developmental event in female embryos, but is a flexible process that tunes RNA synthesis proportionally to the output of the second X allele across developmental states. These new insights alter the model of dosage compensation in early mammalian embryogenesis, bridging the allele-specific dynamics of XCU and XCI.

## Results

### Naïve female embryonic cells lack X-upregulation.
Exit from pluripotency is accompanied by rXCI in females[14]. We modeled this process by differentiating naïve F1 mESCs of mixed genetic background (C57BL6/J × CAST/EiJ) cultured under 2i condition (Gsk3 + MEK inhibition) towards primed epiblast stem cells (EpiSCs; Activin A + FGF2) for up to 7 days, during which cells were captured for scRNA-seq (Fig. 1a and Supplementary Data 1). By using Smart-seq3[15] we obtained gene-copy-specific expression measurements, as full-length read coverage provided the allelic origin of transcripts via parental polymorphisms and unique molecular identifiers (UMIs) yielded original RNA counts (Supplementary Fig. 1a). After quality filtering, we captured the

allelic expression of up to 576 X-linked and 18,043 autosomal genes in 687 deep-sequenced cells, highlighting that our data provided near genome-wide allelic resolution and the sensitivity to study allelic regulation on the gene level in single cells.

The mESC-to-EpiSC transition triggered distinct expression changes accompanied by the loss of pluripotency factors (e.g., Sox2 and Nanog) and induction of lineage-specific factors (e.g., Fgf5 and Krt18) together with related pathways (Fig. 1b, c, Supplementary Fig. 1b, c, and Supplementary Data 2), signifying successful stem cell priming. As expected, female naïve mESCs, carrying two active X alleles (XaXa state), demonstrated an elevated total X-gene expression dosage that diminished upon exit from pluripotency (Fig. 1d). This dose-diminishing effect has previously been attributed to the silencing of one out of two hyperactive X alleles by XCI[7,16]. We confirmed the elevated X dosage in naïve female XX mESCs compared to both male XY and female XO (Turner syndrome) mESCs in bulk RNA-seq[17,18] (Fig. 1e). At the same time, we recorded female XX mESC expression to be less than the twofold higher expected relative to XY and XO if comparing a biallelic hyperactive XaXa state to a single hyperactive X allele in XY/XO cells (median XY = 1.42 and XO = 1.54-fold, relative to XX) (Fig. 1f). To dissect X dosage to allelic resolution in our scRNA-seq data, we stratified female cells according to XCI status into biallelic and monoallelic X-chromosome states (XaXa, and XaXi/XiXa, respectively) (Fig. 1g and Supplementary Fig. 1d) calculated from X-chromosome allelic expression imbalance, indeed confirming decreased total X-linked expression upon XCI (Supplementary Fig. 1e). Surprisingly, when resolved onto the separate alleles, X-linked expression in XaXa cells was on par with autosomal levels for each allele whereas female X-inactive states (XaXi/XiXa) and male (XY) cells exhibited distinct upregulation of the single active X-chromosome copy (Fig. 1h). Remarkably, this implies that XCI does not initiate silencing upon one of two hyperactive X chromosomes, which is distinct from what is widely believed and modeled for the dosage compensation process[3,4,7,16]. Notably, upregulation was observed X-chromosome-wide (Supplementary Fig. 1f) and female XaXa cells consistently lacked XCU regardless of days of EpiSC priming (Supplementary Fig. 1g). This was validated by reanalyzing published 3′ UMI-tagged allele-resolved scRNA-seq data of 2i withdrawal in mESCs[19], under which XCI initiates at a lower rate compared to EpiSC priming[14], allowing us to observe the effect of biallelic versus monoallelic X expression over a wide timespan (Supplementary Fig. 1h–j). Notably, our finding that two moderately expressed X alleles (XaXa state) achieve higher total expression dose than a single hyperactive X allele implies that XCU does not attain full twofold upregulation at the transcript-count level. Indeed, calculating relative gene-wise expression of the same active allele transitioning from XaXa into XaXi state (Fig. 1i), we observed a pronounced shift in X-linked expression yet with fold changes below two (median $X_{C57}$ = 1.62, $P_{C57}$ = 1.41 × 10$^{-40}$ and median $X_{CAST}$ = 1.67, $P_{CAST}$ = 5.72 × 10$^{-48}$, Wilcoxon signed-rank test). Indeed, even as the Xa allele became hyperactive after XCI, the combined output of both basally expressed X alleles in the XaXa state remained higher (Fig. 1j). While these findings conceptually conflict the model that XCI acts on a biallelically hyperactive XaXa state[3,4], the scenario remains compatible with the notion that increased dosage of X-linked pluripotency factors in naïve female XaXa cells delays differentiation of female mESCs[20].

### Two waves of X-upregulation.
Naïve mESCs are derived from, and mimic, the inner cell mass (ICM) of preimplantation embryos. XCU was previously proposed to be present prior to ICM formation[5,11,12] but the lack of hyperactivation in naïve female mESCs that we observed suggested that XCU might be reversed during embryonic development. To map the

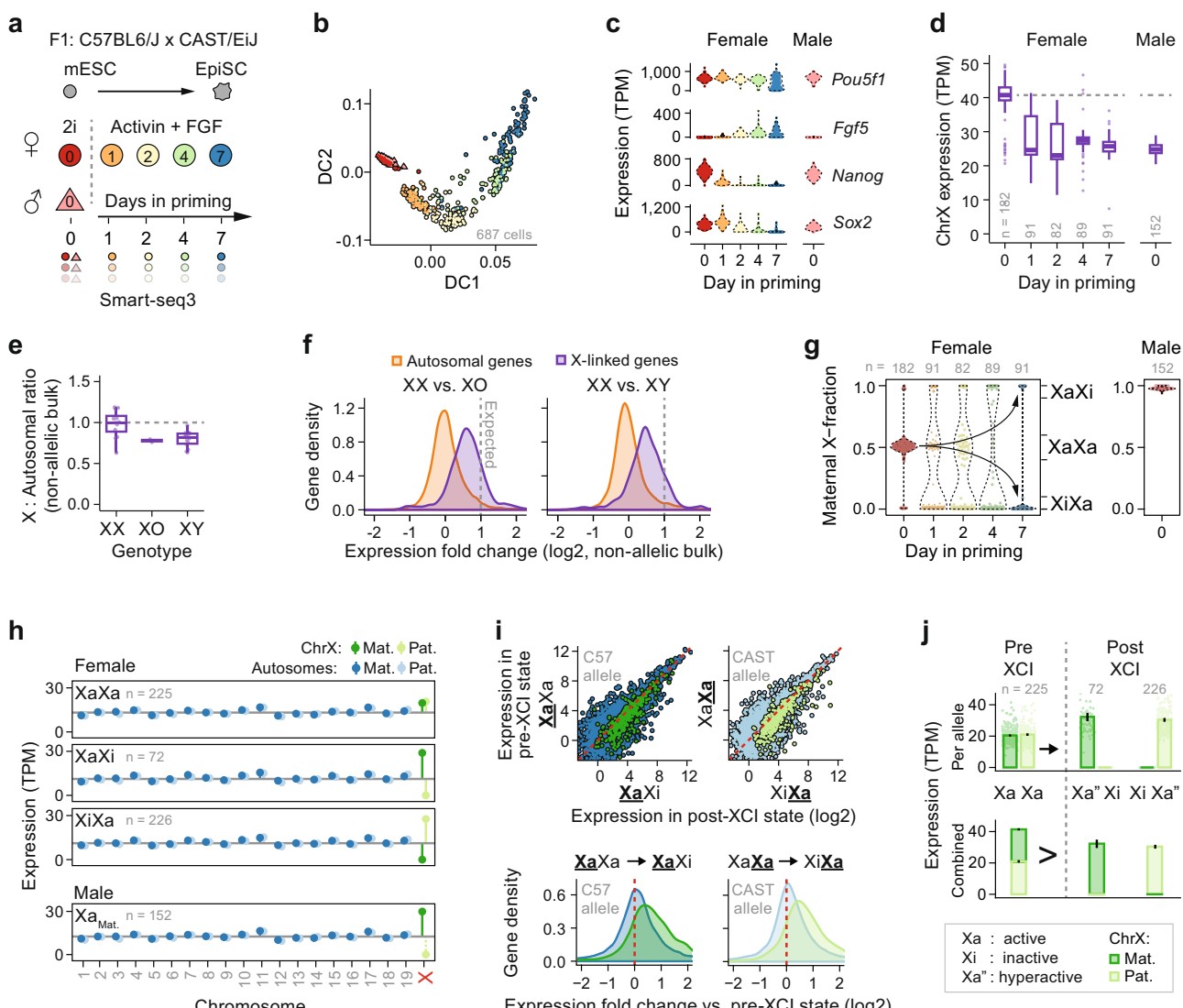

**Fig. 1 The state of two active X chromosomes lack X-upregulation. a** Schematic overview of the experimental setup. C57BL6/J × CAST/EiJ F1 hybrid mESCs were maintained under 2i conditions and female mESCs were primed into EpiSCs using Activin A and FGF2 for up to 7 days to induce X-inactivation (XCI). Cells were collected at conditions and days of priming as indicated and subjected to scRNA-seq using Smart-seq3. Allelic detection was enabled using strain-specific SNPs in the sequencing data. See Supplementary Data 1 for cell annotations. **b** Diffusion map showing the differentiation trajectory of mESC priming towards EpiSCs (visualized using the top 1000 variable genes). Cell colors as denoted in panel **a**. **c** Violin plots of marker gene expression of pluripotency and differentiation markers *Pou5f1*, *Fgf5*, *Nanog*, and *Sox2* along EpiSC priming. **d** Box plots showing total expression of chrX per cell, sex, and timepoint along EpiSC priming ($n = 1158$–$1337$ genes). Data are shown as median, first, and third quartiles, and 1.5x inter-quartile range (IQR). **e** Box plots of X:Autosomal ratios for bulk RNA-seq of mESCs of XX, XO (Turner syndrome), and XY genotypes ($n = 409$–$522$ chrX genes). Bulk RNA-seq libraries: XO ($n = 2$), XY ($n = 14$), XX ($n = 12$), each data points shown as a dot. Genes: autosomal ($n = 10{,}961$), X ($n = 405$). Data are shown as median, first, and third quartiles, and 1.5x IQR. **f** Density plot of gene-wise expression fold changes in bulk RNA-seq from female mESCs carrying two X chromosomes (XX) relative to female mESCs lacking one X-chromosome copy (XO) or male mESCs (XY). Dashed lines indicate expected twofold expression fold change, $n =$ same as in **d**. **g** Violin plots of fraction maternal X-chromosome expression in cells (dots) along EpiSC priming. Female cells were classed according to allelic X-chromosome expression bias, i.e., XCI state, where each X allele is active (Xa), inactive (Xi), or semi-inactive (Xs, not shown) (right y-axis). **h** Allele-resolved expression per cell for all chromosomes grouped by sex and XCI status, for autosomes (blue; $n = 15{,}683$–$16{,}543$ genes) and chrX (green; $n = 508$–$518$ genes), plotted as median ± 95% confidence interval. **i** Expression scatter plots of one allele in XaXa vs. X-inactive state (XaXi or XiXa; C57 and CAST allele active, respectively) in female cells (left) and density plots of expression level fold change of the same allele in XaXi or XiXa state relative to the XaXa state (right). Blue: Autosomal genes ($n = 13{,}696$–$14{,}845$), green: X-linked genes ($n = 383$–$433$). **j** Bar plots of allele-resolved chrX expression ($n = 508$–$518$ genes) in female mESCs. Naïve mESCs with of two active X chromosomes (XaXa) lack XCU (top). As XCI inactivates one allele (Xi) the remaining allele becomes transcriptionally hyperactive (Xa"). Biallelic total RNA output of XaXa exceeds that of the monoallelic Xa" state (bottom). Shown as mean ± 95% confidence interval. **b**, **d**, **g**, **h**, **j** Number of individual cells per condition shown within plots.

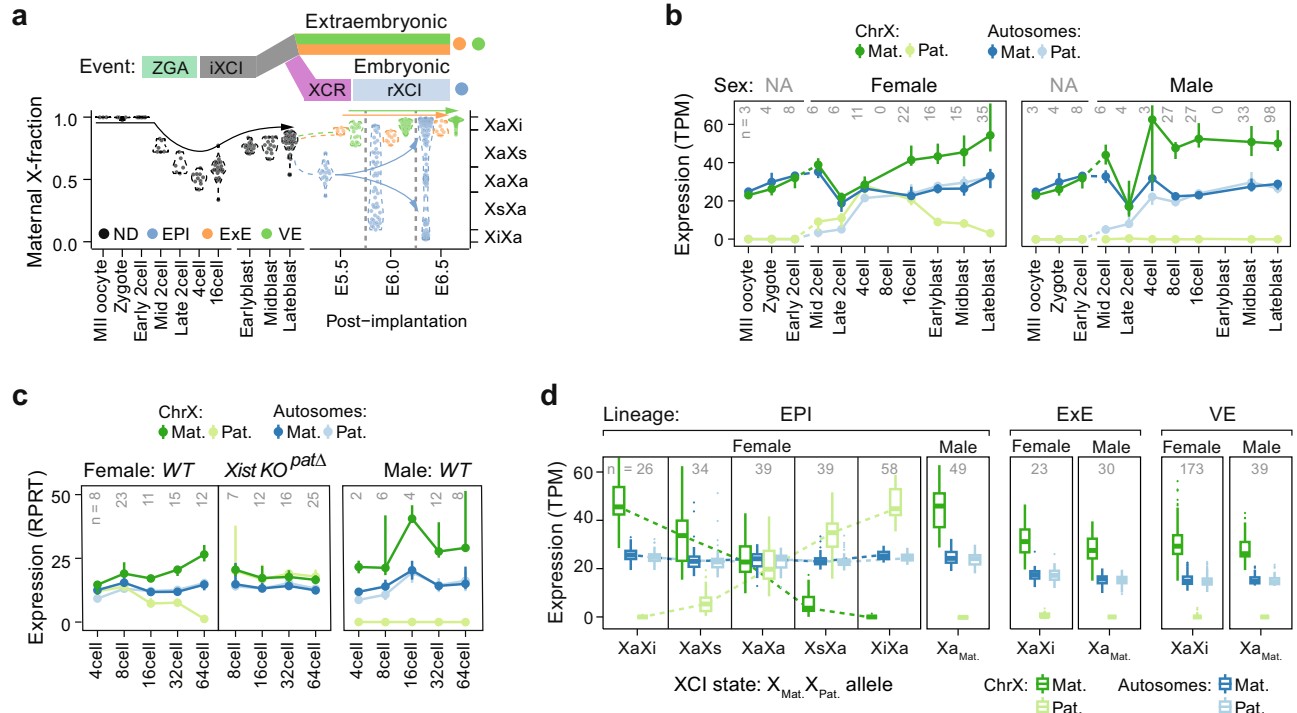

**Fig. 2 Allele-level map of X-upregulation dynamics during early murine embryonic development. a** Violin plots of maternal fraction X-chromosome expression in cells (dots) across the first week of female mouse embryo development (x-axis), with known key events of allelic and X-chromosome regulation indicated above; ZGA (mid-2- to the 4-cell stage), iXCI (initiating at 8- to 16-cell stage), XCR (epiblast-specific, late blastocyst to early implantation), and rXCI (epiblast-specific following XCR). Past ZGA, the XCI states of female cells were inferred according to allelic X-chromosome expression bias (right y-axis). Cell-lineage assignment: not determine (ND), epiblast (EPI), extraembryonic ectoderm (ExE), visceral endoderm (VE). **b** Allele-resolved expression during preimplantation development chrX (green; $n = 323$–437 genes) and autosomes (blue; $n = 11,164$–13,347 genes) in cells of female and male embryos. Data are shown as median ± 95% confidence interval. **c** Allele-resolved expression for wild-type (WT) or female $Xist^{pat\Delta}$ knockout embryo cells along the time window of iXCI establishment for autosomes (blue; 1712–7908 genes) and chrX (green; 36–248). Legend as in panel **d**. RPRT reads per retro-transcribed length per million mapped reads. Data are shown as median ± 95% confidence interval. **d** Box plots of allelic gene expression levels in postimplantation cells grouped by sex and lineage. Female epiblast cells are further grouped by rXCI state where X alleles are assigned as active (Xa), inactive (Xi), or semi-inactive (Xs) i.e., ongoing XCI establishment. Shown for chrX (green; $n = 597$–713) and autosomes (blue; $n = 14,941$–17,643 genes). Data are shown as median, first, and third quartiles, and 1.5x IQR. **b–d** Number of individual cells per condition shown within plots.

X-upregulation dynamics in vivo, we leveraged the large allele-level Smart-seq datasets we recently generated across early murine embryogenesis[14,21,22], covering key developmental time points from the oocyte/zygote up until gastrulation and spanning the known major events of embryonic X-chromosome regulation;[1] i.e., imprinted XCI (iXCI), X-chromosome reactivation (XCR), and rXCI (Fig. 2a). Allelic expression was balanced for X and autosomes in mature (MII) oocytes up until two-cell stages where mRNAs originate only from the maternal genome. However, around the completion of zygotic genome activation (ZGA, ~4-cell stage), where biallelic autosomal transcription is achieved, XCU was specifically observed in male cells (Fig. 2b). Conversely, female four-cell embryos exhibited biallelic XaXa expression that lacked XCU, recapitulating the observations in naïve XaXa mESCs. Maternal-specific XCU was first detected around the 8–16-cell stage and was maintained throughout pre-implantation blastocyst development (Fig. 2c), notably coinciding with the iXCI dynamics on the paternal allele. Because lineage specification commences during blastocyst development, we classified blastocyst cells into epiblast (EPI), trophectoderm (TE), and primitive endoderm (PrE) lineages (Supplementary Fig. 2a, b), revealing XCU to be present in all late-blastocyst lineages, including EPI cells prior to XCR (Supplementary Fig. 2c). Importantly, the sex-specific temporal dynamics of XCU closely followed events that would otherwise result in imbalanced

chromosomal dosage, suggesting that XCU follows external X-chromosome-dosage imbalances. To test this hypothesis, we analyzed XCU in allele-resolved scRNA-seq data from $Xist^{pat\Delta}$ knockout embryos genetically designed to lack iXCI[11]. Indeed, female $Xist^{pat\Delta}$ knockout embryos did not initiate XCU (Fig. 2c), indicating that hyperactivation is initiated as a response to imbalanced dosage.

As cell lineages are transcriptionally distinct in postimplantation embryos[22] (E5.5-6.5), we continued the analyses in a lineage-specific manner. Whereas extraembryonic lineages (visceral endoderm, VE; extraembryonic ectoderm, ExE) retain iXCI in female cells, EPI cells undergo XCR followed by rXCI (Fig. 2a). Strikingly, we found that cells of the extraembryonic lineages maintained XCU along with iXCI (Fig. 2d) whereas female EPI cells residing in reactivated XaXa state (XCR state) lacked XCU regardless of the embryonic age or inferred developmental pseudotime of the cells (Fig. 2d and Supplementary Fig. 2d–g), importantly demonstrating erasure of XCU in vivo. This was followed by a second wave of XCU in cells where rXCI was either underway or completed (Fig. 2d and Supplementary Fig. 2d, f), confirming that XCU regulation is highly dynamic also in vivo.

**A quantitative relationship between X-upregulation and X-inactivation.** Next, we investigated whether XCU responded quantitatively to the lowering of expression from the inactivating

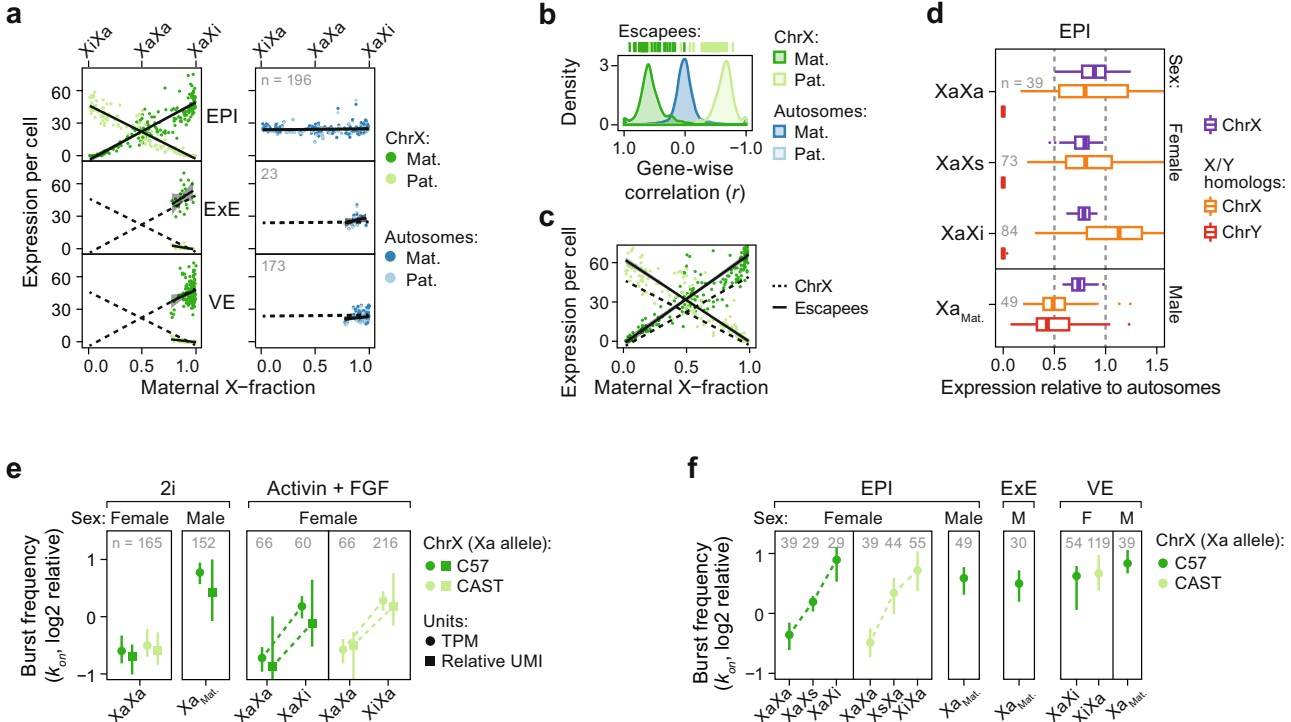

**Fig. 3 X-upregulation follows a tuning-like mode of regulation. a** Scatter plot of expression level per allele (y-axis) and maternal X-fraction per female cell (x-axis) with lines indicating linear model mean ±95% confidence interval. Data are shown for three lineages (EPI, ExE, and VE). Dashed black lines project EPI expression. Shown for autosomes (blue; $n = 14{,}941$–$17{,}643$ genes) and chrX (green; $n = 597$–$713$). **b** Density plot of gene-wise ($n = 712$ genes) correlations (Pearson) against maternal X-fractions in EPI cells ($n = 196$). **c** Same as **a**. but shown separately for all X-linked (ChrX) or XCI escapee expression per cell and allele. **d** X:Autosomal and Y:Autosomal expression ratios per cell shown as box plots grouped by lineage, sex, and XCI state for chrX ($n = 129$–$410$ genes) and ancestral X-Y homologs ($n = 5$–$9$ genes) separately. X alleles indicated as active (Xa), inactive (Xi), or semi-inactive (Xs). Data are shown as median, first, and third quartiles, and 1.5x IQR. **e** Transcriptional burst frequency ($k_{on}$) for active X alleles (Xa per parental strain) grouped by sex, culture condition, and XCI state, shown as median ± 95% confidence interval, inferred by either TPM (dot) or relative UMIs (square). Data are shown relative to median autosomal burst frequency. $N$ genes $= 10{,}836$–$16{,}714$ and $348$–$545$ for autosomes and chrX, respectively. **f** Transcriptional burst frequency ($k_{on}$) of the Xa allele in cells of different lineage, sex, and XCI state in mouse embryogenesis in vivo, shown as median ± 95% confidence interval relative to median autosomal burst frequency, $n = 252$–$316$ genes. X alleles indicated as active (Xa), inactive (Xi), or semi-inactive (Xs). Legend as in panel **f**, female ExE lineage lacked sufficient cells ($n \leq 20$) for kinetic inference. **a**, **d**–**f** Number of individual cells per condition shown within plots.

allele during XCI progression. If *trans*-acting factors gradually shift towards the Xa allele upon XCI, as hypothesized from our previous work[23,24], XCU would not show an on/off pattern but would "tune" expression according to the cellular degree of XCI. To ensure that such tuning-like dynamics could be correctly measured on the allele-level across different levels of allelic imbalance, we constructed and sequenced mock libraries of equimolar concentration containing various spiked ratios of purified C57 and CAST RNA (Methods), indeed demonstrating that allelic ratios could be faithfully captured down to around 100,000 reads per sample or with as few as 100 genes using UMIs as well as read counts (Supplementary Fig. 3a, b).

As rXCI is an asynchronous process[14,22], it represents an ideal system to evaluate the XCU modus. Indeed, as rXCI was established in EPI cells, the other allele displayed corresponding chromosome-wide compensation in accordance with a tuning-like mode (adjusted $R^2_{maternal} = 0.80$, $P_{maternal} = 2.92 \times 10^{-69}$, adjusted $R^2_{paternal} = 0.91$, $P_{paternal} = 1.19 \times 10^{-103}$, linear regression; Fig. 3a). This tuning-like behavior was further confirmed in independent UMI-count data of 2i withdrawal in mESCs[19] (adjusted $R^2_{C57} = 0.84$, adjusted $R^2_{CAST} = 0.86$, $P < 2.2 \times 10^{-16}$, linear regression; Supplementary Fig. 3c). Strikingly, variability of iXCI completeness in extraembryonic lineages and preimplantation stages also followed the trend projected from postimplantation EPI cells (Fig. 3a and Supplementary Fig. 3d), suggesting that

the first iXCI-associated wave of XCU is achieved by the same mechanism as the second rXCI-associated wave. Together, these findings provide evidence for continuous feedback of XCU. Surprisingly, genes known to escape XCI followed similar tuning-like trends as other X-linked genes in female cells ($P_{maternal} = 0.62$, $P_{paternal} = 0.71$, Kolmogorov–Smirnov test; Fig. 3b, c). Furthermore, we observed that the subset of escapee genes having ancestral homologs remaining on the Y chromosome[25] lacked dosage compensation in males whereas the corresponding X-linked homologs displayed XCU in female cells of all lineages (Males: $P > 0.05$, Females: $P_{EPI} = 6.02 \times 10^{-30}$, $P_{VE} = 3.91 \times 10^{-15}$, $P_{ExE} = 4.84 \times 10^{-2}$, one-sample Wilcoxon test, $\mu = 0.5$; Fig. 3d and Supplementary Fig. 3e). Therefore, the combined effect of biallelic expression and gene-specific XCU may explain why escapee genes tend to be expressed at higher levels in females[1] but at the lower output from the inactive X allele[26–28]. How could the tuning-like action of XCU have been overlooked in previous studies? We speculated that this was due to the lack of allele-resolution measurements of dosage compensation[5–9,11,19,29]. Indeed, all approaches we tested on our data utilizing total expression levels (total X expression, X:Autosomal ratios, Female:Male ratios) failed to identify tuning-like XCU as all non-XaXa cell states produced similarly balanced expression (Supplementary Fig. 3f–h), explicitly exposing the risk of inferring allelic processes from nonallelic measurements.

Steady-state mRNA levels are determined by synthesis and degradation. We have previously shown that expression-matched X-linked- and autosomal transcripts have similar decay rates[24] and others have associated steady-state XCU with increased transcriptional initiation[8,9,30], suggesting that XCU is primarily controlled at the level of transcription. Indeed, when dissecting expression levels into kinetic parameters of transcriptional bursting (two-state model [on/off]; burst frequency [$k_{on}$] and burst size [$k_{syn}/k_{off}$]; Supplementary Fig. 3i), we previously found XCU to be driven by increased transcriptional burst frequency[24,31]. To test whether bursting patterns underlie the tuning-like mode of XCU, we inferred allele-level kinetic parameters from our mESCs using molecule- (UMI) and read-count (TPM) Smart-seq3 data (Methods). The two XaXa alleles displayed moderate and balanced burst frequency ($k_{on}$) whereas all states of monoallelic X-chromosome expression (i.e., XCU states) displayed markedly increased burst frequency ($FDR_{TPM}$ ≤$3.97 \times 10^{-10}$, $FDR_{UMI}$ ≤$5.68 \times 10^{-03}$, FDR-corrected Mann–Whitney $U$-test; Fig. 3e) whereas burst size ($k_{syn}/k_{off}$) remained largely unchanged (FDR ≥0.08, Supplementary Fig. 3j). Next, we inferred transcriptional kinetics during rXCI establishment in vivo and indeed found that burst frequency on the Xa allele was progressively increased in a tuning-like manner in epiblasts (Fig. 3f and Supplementary Fig. 3k). Interestingly, the Xi allele lost burst frequency at a higher rate than gained at the Xa allele (Supplementary Fig. 3l), suggesting that residual mRNA molecules help buffer allelic imbalance. Male cells and extra-embryonic cells subject to iXCI similarly displayed elevated burst frequency on the hyperactive X allele (Fig. 3f), pointing towards allelic tuning by transcriptional bursting as a general mechanism of XCU.

**The epigenetic state of X-upregulation.** With our new insights into allele-specific XCU dynamics, we sought to directly contrast the epigenetic state of the hyperactive and non-hyperactive X allele for the first time. To do so at allelic and cellular resolution, we utilized a single-cell multi-modal profiling assay combining Smart-seq3 and scATAC-seq in parallel for the same cell (Supplementary Data 1). We applied this method to our mESC priming model throughout rXCI and XCU establishment (Fig. 4a and Supplementary Fig. 4a), providing the first combined scRNA/ATAC-seq profiling with an allelic resolution to our knowledge. As expected, the mESC-to-EpiSC transition was accompanied by distinct changes in chromatin accessibility of pluripotency and differentiation markers (Fig. 4b, c), matching differential expression in our initial Smart-seq3-only experiment (Odds ratio = 2.66, $P = 8.34 \times 10^{-15}$, Fisher's exact test). Next, we integrated single-cell expression and chromatin accessibility measurements, which further confirmed the agreement between the RNA and DNA modalities (Fig. 4d and Supplementary Fig. 4b). To assess whether the combined scRNA/ATAC-seq assay could accurately detect gene- and allele-specific accessibility, we investigated imprinted autosomal genes[32] which showed skewing to the expected parental alleles in both expression and accessibility (Fig. 4e, f). Next, we grouped cells into different XCI states based on the allelic expression of the RNA modality, which demonstrated concurrent loss of chromatin accessibility on the Xi allele (Fig. 4g) whereas autosomes remained biallelically accessible (Supplementary Fig. 4c). Surprisingly, unlike the distinct shift in X-linked RNA levels, accessibility of the active allele was not notably increased upon XCU neither in male nor female cells (Supplementary Fig. 4d). This was also the case when matching expression and accessibility for the same gene where only RNA-level upregulation was observed relative to the biallelic XaXa state, regardless of the degree of XCU (Supplementary Fig. 4e, f),

indicating that the transcriptional action of XCU transpires on a basal state of chromatin openness. To address this further, we reanalyzed allele-resolved native histone ChIP-seq (H2AK119Ub, H3K27me3, H3K4me1, H3K4me3, H3K27ac, H3K9ac, and H4ac) timeseries data for mESCs with DOX-inducible XCI[33]. In agreement with our combined scRNA/ATAC-seq data, the number of enriched regions for all histone modifications remained constant on the Xa allele throughout XCI/XCU progression (Supplementary Fig. 4g), as did modification density at both promoters and enhancers (Supplementary Fig. 4h). To explore potential spatial patterns of XCU accessibility, we separated alleles based on the degree of XCI completion (per modality) which indeed revealed unchanged accessibility across the Xa allele whereas Xi accessibility was preferentially lost at regions gaining the heterochromatin histone modification H3K27me3 (Fig. 4h). This corroborates our initial finding that the chromatin accessibility landscape remains nearly unchanged upon XCU. Intrigued by these observations at the chromatin level, we hypothesized that the burst-frequency-driven XCU may not be modulated by enhancer state per se but through enhancer-to-gene contacts[34,35]. To explore this, we analyzed allele-resolved time-series in situ Hi-C data for mESCs undergoing differentiation for up to 10 days[36] (Methods). Unlike the Xi allele, that assumes a distinct bipartite mega-domain structure upon XCI[36–38], Xa retained its global long-range chromosome conformation (Supplementary Fig. 5a). However, the shorter-range chromatin domains on Xa became increasingly distinct as the cells underwent XCU (Supplementary Fig. 5b, c), suggesting that XCU is associated with increased chromatin contacts, consistent with the observed increase in transcriptional burst frequency. While this observation is preliminary and warrants further study, we do note that the Xa allele has also been found to be more structured in cells with XCI in two recent studies[39,40], in agreement with our findings on XCU.

## Discussion

In this study, we uncovered unanticipated flexibility of XCU in controlling X-linked expression, with fundamental implications to our understanding of dosage compensation. By tracing expression levels at allelic resolution in single cells during early murine embryonic development, we identified key sex- and lineage-specific events and timing for initiation, maintenance, erasure, and reestablishment of allelic X-upregulation to compensate otherwise imbalanced RNA levels. Notably, we showed that XCU is achieved by transcriptional burst frequency increase as a universal kinetic drive, co-occurring with increased chromatin compartmentalization. In contrast to the tightly controlled initiation of XCI[20], we found that XCU acts in a tuning-like manner as a direct response to imbalanced X dosage, i.e., requiring ZGA or XCI for establishment in XY males and XX females, respectively (Fig. 2b, c). This is distinct from previous reports suggesting XCU be active already in the zygote[5] or progressively after the four-cell stage in both sexes[11,12]. These discrepancies may be explained by the inability of previous nonallelic gene expression measurements to correctly distinguish XCU in the presence of parallel confounding allelic processes such, as ZGA and XCI, whereas our present study directly attributes XCU to the active X allele. Our surprising finding that female cells with biallelic expression (naïve female mESCs, four-cell embryos, and uncommitted epiblasts; XaXa state) lack previously assumed XCU[7,8,41] is in line with observations of balanced X:A dosage in haploid cells (including MII oocytes; Fig. 2b) or other cellular states harboring two active X chromosomes, such as primary oocytes and primordial germ cells[5,29,42,43]. Furthermore, the gene expression dosage of two Xa alleles is known to be incompatible with sustained embryonic development[20,44] which fits with the observation of

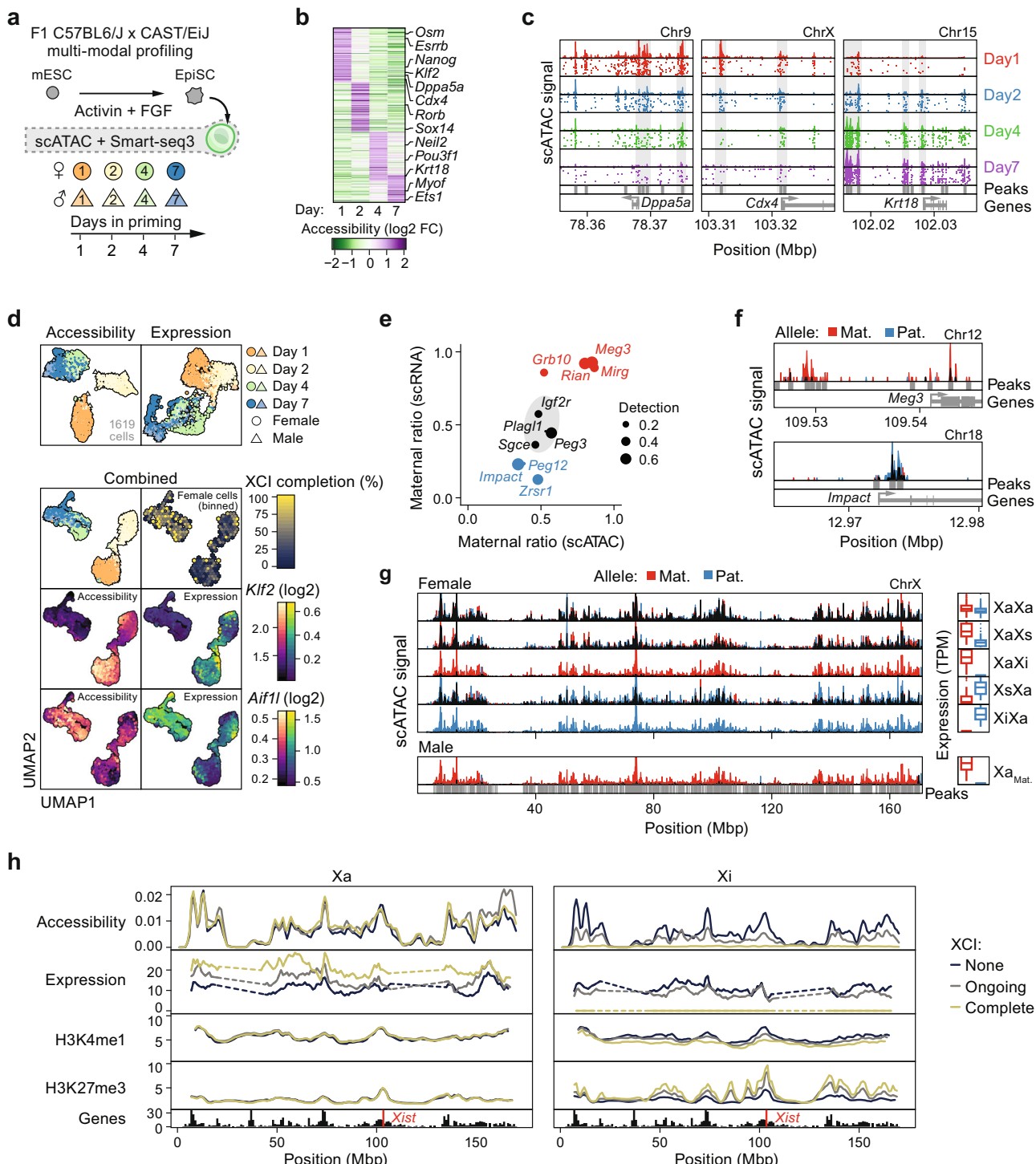

incomplete compensation by XCU at the RNA level (~1.6-fold) reported by us and others[5–9,24,30] (Fig. 1). Using allele-resolved chromatin analyses, including joint scRNA/ATAC-seq, we characterized epigenetic features of XCU. In contrast to readily observable differentiation- and XCI-related changes, we found the broad chromatin landscape of XCU to be unaltered which, in combination with the increased transcriptional burst frequency upon XCU, may explain why active histone modifications do not scale proportionally with RNA levels on the hyperactive X allele[9,30].

These revelations prompt a revised model of the sequence of events by which mammalian X-chromosome dosage compensation is achieved in the presence of other allelic processes during

development (i.e., ZGA, iXCI, XCR, and rXCI; Fig. 5a). This conflicts with the widely-held belief that XCI acts on hyperactive Xa alleles in females[7,8,41] ("Default XCU" model; Fig. 5b top). Instead, our data fit a model where the two Xa alleles are moderately expressed and that XCU gradually tunes Xa expression levels throughout XCI proportional to dosage imbalances ("Elastic XCU" model; Fig. 5b bottom). The transcription-driven XCU we observe may act on top of other layers of regulation at post-transcriptional- and translational levels[1], that could be now be investigated in detail as the XCU timing and dynamics are mapped. Although our current data do not pinpoint the upstream regulatory events and elements of the burst-frequency-driven

**Fig. 4 Tracking epigenetic features of the X chromosome upon upregulation establishment. a** Schematic overview of the experimental setup. C57BL6/J × CAST/EiJ F1 hybrid male and female mESCs were primed into EpiSCs using Activin A and FGF2 for up to 7 days to induce XCI. Cells were collected at conditions and days of priming as indicated and subjected to single-cell multi-modal profiling of expression (Smart-seq3) and chromatin accessibility (scATAC). See Supplementary Data 1 for cell annotations. **b** Heatmap of accessibility changes along EpiSC priming. Acc. gene accessibility score. **c** Genome tracks of representative genes changing accessibility along EpiSC priming. Shown as single-cell accessibility and merged pseudobulk tracks. **d** UMAP dimensionality reductions for accessibility (top 25,000 features), expression (top 1000 HVGs), or the combined dimensions per cell ($n = 1619$). For combined coordinates, cells are also shown colored based on binned XCI completion (female cells only), or accessibility/expression levels of *Klf2* and *Aif1l*. **e** Maternal ratios for imprinted genes based in scRNA-seq (y-axis) and scATAC-seq (x-axis). Detection detected in fraction of cells. The center ellipse indicates 95% confidence interval for all genes. **f** Genomic tracks of allelic pseudobulk accessibility for representative genes from **e**. **g** Genomic tracks of allelic pseudobulk accessibility for the entire X chromosome. Cells grouped based on XCI state and sex with the corresponding expression of chrX shown as box plots to the right. X alleles indicated as active (Xa), inactive (Xi), or semi-inactive (Xs) based on X-chromosome allelic expression bias. **h** Rolling average of accessibility and expression (TPM) across each X allele, grouped based on the degree of XCI completion (None = 0–20%, Ongoing = 20–80%, and Complete = 80–100% allelic bias). Allelic native ChIP-seq shown for permissive H3K4me1 and repressive H3K27me3 histone modifications (where None = 0 h, Ongoing = 12 h, and Complete = 24 h DOX-induced XCI). Location of *Xist* indicated in red.

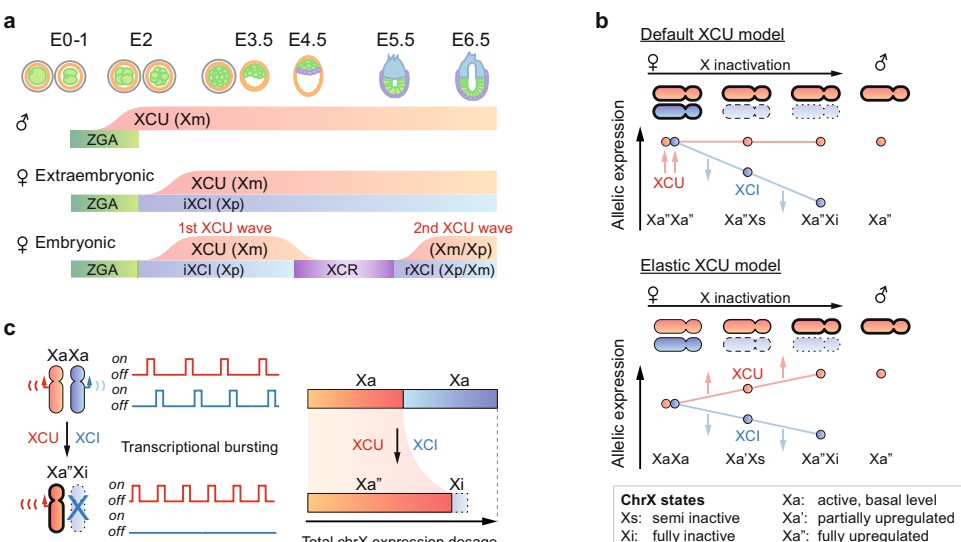

**Fig. 5 A unified model of dosage compensation by X-upregulation and inactivation. a** Overview of X-upregulation (XCU) timing and dynamics across mouse pre- and early postimplantation development in the two sexes and embryonic/extraembryonic lineages. zygotic genome activation (ZGA), imprinted X-chromosome inactivation (iXCI), X-chromosome reactivation (XCR), random X-chromosome inactivation (rXCI). **b** Previous model of dosage compensation ("Default XCU"), proposing that X-upregulation (XCU) is already present in female XaXa state after which X-inactivation (XCI) operates on one allele achieving balanced X-chromosome expression levels. Our allele-level data of XCU introduce a different model ("Elastic XCU") in which XCU is established and tuned in relationship to the degree of XCI on the second allele. **c** As one allele is gradually silenced (Xi) in female cells, transcriptional burst frequency is increased on the second allele (Xa), making it hyperactive (Xa"). However, the single hyperactive allele (Xa") does not fully reach the combined expression level of the two moderately active alleles in the female XaXa state.

XCU, it is worthwhile to hypothesize on their nature. Our findings hint that increased transcription factor concentrations at the hyperactive Xa allele could play a role in its regulation. Not only would the single X allele in XY males and XO females be subjected to higher doses of factors transcribed from diploid autosomes, but increased chromatin contacts might also increase local transcription factor concentrations through loop-mediated trapping[45]. As transcriptional burst frequency is controlled by both local factor concentrations[46] and enhancer-promoter contacts[31,34,35], this could result in the increased transcriptional burst frequency in XCU we observe (Fig. 3e, f). Furthermore, the same model can operate for XCI in XX females as transcription factors are rapidly excluded from the Xi allele when repressive compartments are established[47]. As XCI is a gradual process[17], factor complexes could progressively shift to the Xa allele in line with the tuning-like mode of XCU. Collectively, this would result in balanced RNA levels, explaining how the total dosage of X-encoded transcripts are maintained so surprisingly stable throughout development and XCI progression in mammalian species[5,7,10,11,48] and why XCI escapees are expressed at higher

levels from the Xa allele[26–28] whereas X-Y homologs are not dosage compensated in males as both the X- and Y-chromosome is active (Fig. 3d and Supplementary Fig. 3e). As two active X chromosomes are considered a hallmark of the naïve female stem cell state and a gold standard in reprogramming studies[49], our finding that the biallelic XaXa state lacks XCU alters the interpretation of X-chromosome expression level measurements for assessment of reprogramming success and naïveness.

In conclusion, our study exposes X-upregulation as a remarkably elastic process that tunes RNA dosage throughout development, leading to a refined model of mammalian dosage compensation by unifying the allele-specific dynamics of X-inactivation and X-upregulation in achieving balanced transcript-copy numbers in the cell.

## Methods

**Animal housing and ethics statement**. Mice were housed in specific pathogen-free at Comparative Medicine Biomedicum (KM-B), Karolinska Institutet, according to Swedish national regulations for laboratory animal work food and water ad libitum, cage enrichment, and 12 h light and dark cycles. All animal

experimental procedures were performed in accordance with Karolinska Institutet's guidelines and approved by the Swedish Board of Agriculture (permits 17956-2018 and 18729-2019 Jordbruksverket).

**Derivation and culturing of cell lines.** Male and female cell lines were established as previously described in ref. [14]. In brief, mESCs were derived from E4 blastocysts of F1 embryos (female C57BL/6 J × male CAST/EiJ) and adapted to 2i condition by growing them in gelatin-coated flasks in N2B27 medium (50% neurobasal medium [Gibco], 50% DMEM/F12 [Gibco], 2 mM L-glutamine [Gibco], 0.1 mM β-mercaptoethanol, NDiff Neuro-2 supplement [Millipore], B27 serum-free supplement [Gibco]) supplemented with 1000 units/mL LIF, 3 μM Gsk3 inhibitor CT-99021, 1 μM MEK inhibitor PD0325901, and passaged with accutase [Gibco]. To induce differentiation toward EpiSCs, mESCs grown in serum/LIF were plated on FBS coated tissue culture plates (coated overnight at 37 °C) in N2B27 medium supplemented with 8 ng/mL Fgf2 (R&D) and 20 ng/mL Activin A (R&D) at a cell density of $1 \times 10^4$ cells/cm$^2$ and cultured for up to 7 days. Cells were split and replated in the same condition after 1, 2, 4, and 7 days of differentiation. At each split, an aliquot of cells was collected for single-cell sorting into 96-well plates containing Smart-seq3 lysis buffer.

**Single-cell RNA sequencing (Smart-seq3).** scRNA-seq libraries were constructed as previously described in ref. [15] with slight modification. Briefly, cells were single-cell sorted into 96-well low-bind PCR-plates [Eppendorf] containing 3 μl of lysis buffer (0.5 units/μl RNase inhibitor [Takara], 0.15% Triton X-100 [Sigma], 0.5 mM (each) dNTP [Thermo Scientific], 1 μM oligo-dT primer [5′-biotin-ACGAG-CATCAGCAGCATACGAT30VN-3′; IDT], 5% PEG [Sigma]). Sorting was performed using an SH800 [Sony]. Plates were briefly centrifuged immediately after sorting, sealed, and stored at −80 °C. For cell lysis and RNA denaturation, plates were incubated at 72 °C for 10 min and immediately placed on ice. Next, 5 μl of reverse transcription mix (50 mM Tris-HCl, pH 8.3 [Sigma], 75 mM NaCl [Ambion], 1 mM GTP [Thermo Scientific], 3 mM MgCl$_2$ [Ambion], 10 mM DTT [Thermo Scientific], 1 units/μl RNase inhibitor [Takara], 2 μM of template-switch oligo [5′-biotin-AGAGACAGATTGCGCAATGNNNNNNNrGrGrG-3′; IDT], and 2 U/μl of Maxima H-minus reverse transcriptase [Thermo Scientific]) was added to each sample. Reverse transcription was carried out at 42 °C for 90 min followed by ten cycles of 50 °C for 2 min and 42 °C for 2 min and the reaction was terminated at 85 °C for 5 min. PCR pre-amplification was performed directly after reverse transcription by adding 17 μl of PCR mix (containing DNA polymerase, forward and reverse primer) bringing the final concentration in the 25 μl reaction to 1x KAPA HiFi ReadyMix [Roche] 0.1 μM forward primer [5′-TCGTCGGCAGCGTCAGATGTGTATAAGAGACAGATTGCGCAATG-3′; IDT] and 0.1 μM reverse primer [5′-ACGAGCATCAGCAGCATACGA-3′; IDT]). Thermocycling was performed as follows: 3 min at 98 °C, 22 cycles of 20 s at 98 °C, 30 s at 65 °C and 6 min at 72 °C, and final elongation at 6 min at 72 °C. After PCR pre-amplification, samples were purified with AMpure XP beads [Beckman Coulter] at a volume ratio of 0.8:1. Library size distributions were monitored using high-sensitivity DNA chips (Agilent Bioanalyzer 2100) and cDNA concentrations were quantified using the Quant-iT PicoGreen dsDNA Assay Kit [Thermo Scientific]. cDNA was subsequently diluted to 100–200 pg/μl.

Tagmentation was performed using in-house produced Tn5[50]. Two nanograms of cDNA in 5 μl water was mixed with 15 μl tagmentation mix (0.2 μl Tn5, 2 μl 10x TAPS MgCl$_2$ Tagmentation buffer; 5 μl 40% PEG-8000; 7.8 μl water, per reaction) and incubated 8 min at 55 °C in a thermal cycler. Tn5 was inactivated and released from the DNA by the addition of 4 μl 0.2% SDS and 5 min incubation at room temperature. Library amplification was performed by adding 5 μl mix of 1 μM of forward and reverse custom-designed Nextera index primers [forward: 5′-CAAG CAGAAGACGGCATACGAGATNNNNNNNNNNNGTCTCGTGGGCTCGG-3′, reverse: 5′-AATGATACGGCGACCACCGAGATCTACACNNNNNNNNNN TCGTCGGCAGCGTCIDT-3′, where N represents the 10-bp index bases; IDT] and 15 μl PCR mix (1 μl KAPA HiFi DNA polymerase [Roche]; 10 μl 5× KAPA HiFi buffer; 1.5 μl 10 mM dNTPs; 3.5 μl H$_2$O, per reaction), and thermal cycling: 3 min 72 °C, 30 s 95 °C, 13 cycles of 10 s 95 °C; 30 s 55 °C; 30 s 72 °C, followed by final elongation 5 min 72 °C; 4 °C hold. DNA sequencing libraries were purified using 0.8:1 volume of AMPure XP beads [Beckman Coulter]. Libraries were sequenced using NextSeq 550 and High Output kits [Illumina].

**Single-cell joint accessibility and RNA expression (scATAC + Smart-seq3).** The joint single-cell ATAC and Smart-Seq3 analysis was performed as in an early version of Smart3-ATAC[51]. Briefly, single cells were FACS sorted into 384 well plates containing 3 μl lysis buffer (0.03 ul 1 M Tris-pH7.4, 0.0078 μl 5 M NaCl, 0.075 μl 10% IGEPAL, 0.075 μl RNase Inhibitor, 0.075 μl 1:1.2 M ERCC, 2.7372 μl H$_2$O). Immediately after sorting, plates were centrifuged at $1800 \times g$ 4 °C 5 min, placed on ice 5 min, vortexed 3000 RPM 3 min, and centrifuged again at $1800 \times g$ 4 °C 5 min to lyse the cells and spin down the nucleus. Two microliters of the supernatant was then carefully moved to a new 384 well plate for Smart-seq3 mRNA library preparation and the nucleus remained in the original well for scATAC library preparation. The scATAC in situ tagmentation was performed with 2 μl of Tn5 tagmentation mix (0.06 μl 1 M Tris-pH 8.0, 0.0405 μl 1 M MgCl$_2$, 2 μl Tn5) and incubating at 37 °C for 30 min. After the tagmentation, 2 μl of the

supernatant was aspirated and the nuclei were then washed once with 10 μl ice-cold washing buffer (0.1 μl 1 M Tris-pH7.4, 0.02 μl 5 M NaCl, 0.03 μl 1 M MgCl$_2$, 9.85 μl H$_2$O). The remaining Tn5 was inactivated by the addition of 2 μl 0.2% SDS-Triton X-100 followed by incubation at room temperature for 15 min and 55 °C for 10 min. Barcoding PCR was done by KAPA HiFi PCR Kit [Roche] in a final 25 μl reaction (11.5 μl H$_2$O, 5 μl 5X reaction buffer, 0.75 μl dNTP, 0.5 μl KAPA HiFi DNA Polymerase, 2 μl barcoding primers). The thermal cycling program was 15 min 72 °C, 45 s 95 °C, 22 cycles of 15 s 98 °C; 30 s 67 °C; 1 min 72 °C; followed by final elongation at 5 min 72 °C; 4 °C hold. After PCR, 2 μl of each well was pooled and cleaned-up twice using AMPure XP beads (at 1.3X volume).

Smart-seq3 was performed as described above but with 28 PCR cycles to amplify cDNA. Libraries were sequenced as described above on a NextSeq 550.

**Allelic RNA dilution series.** Liver tissue was isolated from 12-week-old male C57BL6/J and CAST/EiJ mice, dissected into 1–2 mm$^2$-wide samples, and transferred to 1 ml TRIzol isolation reagent (Thermo Scientific). The samples were thoroughly homogenized in TRIzol using a metallic tissue grinder before proceeding to RNA extraction. RNA extraction was performed according to the manufacturer's instructions. In brief, 0.2 ml chloroform was added to 1 ml of TRIzol and the samples were vigorously shaken. The samples were incubated at room temperature and the RNA-containing upper aqueous phase was isolated and precipitated with 0.5 ml isopropyl alcohol. The samples were incubated at room temperature for 10 min and centrifuged at $12,000 \times g$ at 4 °C for 10 min. The pellet was washed once with 75% ethanol and the RNA pellet was air-dried for 10 min. The RNA pellets were resuspended in 50 μl of RNase-free water and incubated in a heat block at 55–60 °C for 15 min before measuring the concentration using a Nanodrop 2000 spectrophotometer. RNA from pure C57BL6/J and CAST/EiJ strains was combined at varying ratios (0, 12.5, 25, 37.5, 50, 62.5, 75, 87.5, and 100% C57) for a total of 200 pg RNA which was subjected to Smart-seq3 with slight modification from above. Briefly, tagmentation was done using 0.1 μl Nextera XT ATM Tn5 [Illumina] for 10 min, index primers were used as 0.2 μM each and a 0.6:1 AMPure XP bead ratio was used for cleanup.

**Smart-seq3 data analysis.** A hybrid mouse genome index was constructed by N-masking the reference genome (GRCm38_68) for CAST/EiJ SNPs from the Mouse Genomes Project (mgp.v5.merged.snps_all.dbSNP142)[52] using SNPsplit (0.3.2)[53]. Raw Smart-seq3 data was processed using zUMIs (2.8.0)[54]. Briefly, sample barcodes were filtered and data was aligned with STAR (2.7.2a)[55] [options: --clip3pAdapterSeq CTGTCTCTTATACACATCT] and reads were assigned to both intron and exon features (Mus_musculus.GRCm38.97.chr.gtf) using FeatureCounts[56]. Next, barcodes were collapsed using 1 hamming distance and gene expression was calculated for both reads and UMIs. Finally, allele-level expression was calculated from the zUMIs output as previously described (github.com/sandberg-lab/Smart-seq3/tree/master/allele_level_expression)[15]. Seventy-seven cells were excluded due to low read depth (>3 MADs) in the original experiments and 486 cells were excluded due to low read depth in the multi-omics experiments.

**Processing of published C57BL/6J × CAST/EiJ scRNA-seq data.** Preprocessed Smart-seq data for MII oocyte—16-cell stages were obtained[21] and two cells were excluded (8cell_8-3 and 16cell_4-2) due to potential sample problems (high paternal X ratio in male and non-ZGA with clustering together with Zygote samples, respectively). Preprocessed blastocyst Smart-seq2 data were obtained[21] and 18 late blastocysts were excluded due to low read depth (>3 MADs). Preprocessed Smart-seq2 data for postimplantation embryos was obtained[14,22].

**Genes escaping X-inactivation and X-Y homolog gene lists.** A list of known mouse escapee genes (1810030O07Rik, 2010000I03Rik, 2010308F09Rik, 2610029G23Rik, 5530601H04Rik, 6720401G13Rik, Abcd1, Araf, Atp6ap2, BC022960, Bgn, Car5b, D330035K16Rik, D930009K15Rik, Ddx3x, Eif1ax, Eif2s3x, Fam50a, Flna, Ftsj1, Fundc1, Gdi1, Gemin8, Gpkow, Huwe1, Idh3g, Igbp1, Ikbkg, Kdm5c, Kdm6a, Lamp2, Maged1, Mbtps2, Med14, Mid1, Mmgt1, Mpp1, Msl3, Ndufb11, Nkap, Ogt, Pbdc1, Pdha1, Prdx4, Rbm3, Renbp, Sh3bgrl, Shroom4, Sms, Suv39h1, Syap1, Tbc1d25, Timp1, Trap1a, Uba1, Usp9x, Utp14a, Uxt, Xist, and Yipf6) was compiled from previous work[11,27,32,57] and excluded from analyses where specified.

A list of ancestral X-Y homologs was obtained[25] and sequence similarity of expressed X-Y homolog transcripts was calculated using the nucleotide BLAST webservice[58] using default settings (performed 2020-03-24). The ancestral homologs had an average sequence similarity of 83% and were detected in a maximum of 2.5% female cells whereas maximum male-specific detection was 98.3%, indicating that X-Y homologs show correct sex-specific mapping.

**Expression calculations.** Allelic reference ratios were calculated as Counts$_{C57}$/Counts$_{Total}$ after exclusion of the top 10% expressed X-linked genes to avoid bias from highly expressed X genes. XCI status was determined as active (Xa/Xa), semi-inactivated (Xa/Xs) or fully inactivated (Xa/Xi) for allelic ratios in the intervals (0.4, 0.6), [0.6, 0.9), and [0.9, ∞) and the inverse, respectively. Due to differences in X-controlling elements (Xce) between C57 and CAST strains, the C57 allele is preferentially inactivated during random XCI[32,59–61] which is observable in our data as the number of Xa$_{C57}$Xi$_{CAST}$ vs. Xi$_{C57}$Xa$_{CAST}$ cells. TPM was calculated for gene $i$ in cell

$j$ as $TPM_{ij} = (FPKM_i / \sum_j FPKM_j) \times 10^6$ and allelic TPM was calculated by scaling TPM by reference ratios per gene and cell. Relative UMI counts were calculated relative to total UMIs per sample (percentage of total counts). A gene was considered expressed in a dataset/lineage if the average TPM expression was >0. To obtain a robust estimate for cell-level expression, 20% trimmed means was calculated per cell.

**Expression ratios**. Chromosome:Autosomal expression ratios were calculated for expressed genes (>1 TPM)[7,8,30] as relative to median of autosomes after excluding genes that escape XCI. Additionally, a bootstrapping method[11] was used to account for a different number of genes between chromosomes. For bootstrapped ratios, random autosomal gene sets of the same size as the test set were selected as a background, repeated $n = 10^3$ times.

Female:male ratios were calculated after exclusion of X escapees as TPM relative to gene average per embryonic day and lineage, as well as for the active X allele for allelic data.

**Differential expression of scRNA-seq data**. For Smart-seq3 data, global count data was size-factor normalized using scater (1.12.2)/scran (1.12.1)[62] and genes expressed in >10% of cells were kept. Differential expression was calculated along days of differentiation as a continuous variable using likelihood ratio tests as implemented in MAST (1.10.0)[63]. Gene set enrichment for mouse GO biological process gene sets obtained from MGI (http://www.informatics.jax.org/downloads/reports/index.html#go; accessed 2020-06-23) was performed on the differential expression model against a bootstrapped ($n = 100$) control model as implemented in MAST.

**Dimensionality reduction, clustering, and trajectory inference**. For Smart-seq3 data, highly variable genes (HVGs) were identified from size-factor-normalized counts using scater/scran and ordered by biological variance and FDR. Data was visualized for the top 1000 HVGs using diffusion maps[64].

For in vivo blastocyst data, HVGs were obtained as explained above and dimensionality reduction was performed for top 1000 HVGs using UMAP[65] and cells were Louvain clustered based on top 1000 HVG ranks using scran.

For in vivo postimplantation data, pseudotime trajectories were inferred using Slingshot (1.2.0)[66] from normalized counts following Mclust (5.4.5)[67] clustering on diffusion map coordinates.

**Kinetics inference**. Missing allelic data points were set to 0 if the gene was detected on the other allele. Kinetic parameters were calculated per lineage/genotype/XCI status/growth condition (depending on dataset) using txburst (github.com/sandberg-lab/txburst)[31]. Genes not passing filtering steps were excluded and relative burst frequency ($k_{on}$) and burst size ($k_{syn}/k_{off}$) was calculated relative to the median of expressed autosomal genes (per lineage). Only groups with >20 cells were kept to increase the reliability of the statistical inference.

**scATAC data analysis**. Raw fastq files were tagged with cell names using GNU sed (4.4), quality and adapter trimmed using fastp (0.20.0)[68], and aligned to the N-masked reference genome using bowtie2 (2.4.1)[69] [options: --very-sensitive -N 1 -X 2000 -k 10]. Duplicates were marked using biobambam2 (2.0.87; gitlab.com/german.tischler/biobambam2) and data were merged and split into respective alleles using SNPsplit (0.3.2)[53] [options: --paired --no_sort]. Downstream processing and analysis was performed using ArchR (1.0.1)[70]. Briefly, non-duplicate properly paired and mapped primary alignments were loaded and mitochondrial and Y chromosomes were excluded. Nonallelic data were filtered for cells with at least 1000 fragments and a minimum TSS enrichment of four and predicted doublets were excluded for a total of 437 excluded cells. Enriched peaks were called grouped by differentiation day using the ArchR implementation of Macs2[71]. Expression data were integrated using Smart-seq3 UMI counts, and dimensionality reduction was performed using the ArchR LSI implementation (default parameters for scATAC and top 1000 variable features based on the variance-to-mean ratio for scRNA) and visualized using UMAP with 15 nearest neighbors. Accessibility X:A ratios was calculated as $mean_{chrX}/mean_{Autosome}$ on ArchR gene scores to account for the binary nature of scATAC-seq. Allelic data was filtered to match the cells in the nonallelic data and allelic count matrices were recalculated. Allelic ratios per cell was calculated using the paired expression data, as described above, and X-inactivation completion was calculated per modality as $|0.5-(Counts_{C57}/Counts_{Total})|/0.5$. A list of known imprinted mouse genes (Sgce, Peg3, Peg12, Plagl1, Zrsr1, Peg13, Airn, Impact, Nckap5, Dlx5, Gm5422, Grb10, Meg3, Rian, Mirg, Igf2r, Igf2, and H19) was obtained[32] and only genes detected by both scRNA-seq and scATAC-seq was kept for analysis.

**Re-analysis of published data**. Raw bulk RNA-seq data including both male and female mESCs was obtained[18,72] and quantified to protein-coding transcripts from GENCODE vM22[73] using pseudoalignment with Salmon (0.14.1)[74]. Transcript abundance estimates were summarized to gene-level using tximport (1.12.3)[75] and differential expression was calculated using likelihood ratio tests in DESeq2 (1.24.0)[76]. A full model including cellular genotype (XX, XY, or XO), cell culture condition (2i or serum) and study accession was tested against a reduced model without the genotype term. Lowly expressed genes (<100 average normalized counts) were excluded from the final plots. Serum growth conditions only showed a minor effect on global X expression compared to 2i (not shown), consistent with a low degree of cells exhibiting complete XCI[14].

Preprocessed scRNA-seq data for wild-type and $Xist^{pat\Delta}$ knockout embryos was obtained;[11] processed data from GSE80810. Briefly, allelic reference ratios were used to split normalized expression values (RPRT reads per retro-transcribed length per million mapped reads) into alleles and average expression per cell was calculated using trimmed means for expressed genes (average total RPRT >0) as described above.

Preprocessed scRNA-seq data for mESCs adapting to serum/LIF conditions were obtained;[19] processed data from GSE151009. Briefly, Total UMI count matrices were size-factor normalized as described above and allelic UMI count matrices were used as is. Reference ratios and XCI status was calculated as described above.

Raw native ChIP-seq data for hybrid mESCs was obtained[33], quality and adapter trimmed using fastp (0.20.0)[68], aligned to a CAST/EiJ N-masked genome using Bowtie2 (2.4.1)[69] [options: -N 1] and aligned data was split to respective alleles using SNPsplit (0.3.2)[53]. Peaks were called against Input samples as controls using MACS2 (2.1.2)[71] [options: --broad --broad-cutof 0.01 -f BAMPE -g 2652783500], peaks overlapping ENCODE v2 problematic regions[77] were excluded and consensus peaks were defined as peaks with a reciprocal overlap of at least 25% of peak width between replicates. Normalized coverage was calculated using deepTools (3.3.0)[78] [options bamCoverage --normalizeUsing RPGC --effectiveGenomeSize 2652783500 --skipNAs --ignoreDuplicates --centerReads --blackListFileName mm10-blacklist.v2.ENSEMBL.bed.gz] and profiles were calculated using deepTools for chrX TSSs (Mus_musculus.GRCm38.97.chr.gtf) [options: computeMatrix reference-point --upstream 5000 --downstream 5000 --skipZeros --nanAfterEnd] or chrX enhancers (intersection between H3K4me1 and H3K27ac consensus peaks excluding peaks withing 1 kb of a TSS) [options: computeMatrix reference-point --referencePoint center --upstream 5000 --downstream 5000 --skipZeros] and resulting coverage profiles were normalized against Input.

Raw in situ Hi-C data for hybrid mESCs was obtained[36]. An N-masked dual hybrid mouse genome index was constructed for 129S1/svImJ × CAST/EiJ (mgp.v5.merged.snps_all.dbSNP142) using SNPsplit (0.3.2)[53] and further in silico MboI digested for downstream tools. Data were aligned using HiCUP (0.8.0) [options: --bowtie2 --shortest 50 --longest 700 --digest --zip][79] then split into alleles using SNPsplit, converted to juicer format using HiCUP (hicup2juicer), and reads uniquely corresponding to either allele (requiring at least one mate to map to the genome) were merged. Hi-C contact maps were generated using juicer tools pre (1.22.01) [options: -d -f -q 10 -r 1000000][80] and KR-normalized 1 Mb matrices were extracted using juicer tools (dump observed/Pearsons/eigenvector). Observed contact counts were further normalized for sequencing depth (per million) per chromosome and timepoint.

**Statistics and data visualization**. All statistical tests were performed in R (3.6.1) as two-tailed unless otherwise stated. Heatmaps were visualized using Complex-Heatmap (2.0.0)[81] and all other plots were made using ggplot2 (3.2.1)[82]. Box plots are presented as median, first and third quartiles, and 1.5x inter-quartile range (IQR). For median ± confidence interval plots, bootstrapped 95% confidence intervals ($n = 1000$) were calculated using the percentile method[83] as implemented in the R boot package (1.3-23).

**Reporting Summary**. Further information on research design is available in the Nature Research Reporting Summary linked to this article.

## Data availability
Raw and preprocessed data generated in this study have been deposited in ArrayExpress database under accession codes E-MTAB-9324 (Smart-seq3), E-MTAB-10709 (Allelic dilution series), and E-MTAB-10714 (Combined Smart-seq3+scATAC). Previously published raw data is available at Gene Expression Omnibus under accessions GSE45719, GSE74155, GSE109071, GSE116480, GSE23943, GSE80810, GSE90516, GSE116649, and GSE151009. Data generated during this study are available at github: github.com/reiniuslab/Lentini_XCU_in_vivo.

## Code availability
Code generated during this study are available at github: github.com/reiniuslab/Lentini_XCU_in_vivo.

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

## Acknowledgements

This study was made possible by grants from the Swedish Research Council (2017-01723), the Ragnar Söderberg Foundation (M16/17), the Knut and Alice Wallenberg Foundation (2021.0142), and SRA Stem Cells and Regenerative Medicine (Karolinska Institutet) to B.R. M.E. is supported by The Swedish Cancer Society, The Swedish Childhood Cancer Fund, Radiumhemmets Forskningsfonder, SFO StratRegen, The Swedish Research Council (2020-02940) and Cancer Research KI. A.L. is supported by a postdoctoral fellowship from the Swedish Society for Medical Research. We thank members of the Reinius lab and Colm Nestor for comments on the manuscript, Christoph Ziegenhain for support related zUMIs, and Björn Högberg and Ana Teixeira for use of an Illumina sequencer.

## Author contributions

A.L.: Investigation, methodology, formal analysis, conceptualization, visualization, data curation, writing —original draft, Writing—review and editing. H.C.: Investigation. J.C.N., N.P., C.C., and N.A.: Investigation. M.E. and Q.D.: Methodology and resources. B.R.: Investigation, methodology, conceptualization, supervision, resources, project administration, funding acquisition, writing—original draft, writing—review and editing.

## Funding

## Competing interests

The authors declare no competing interests.
