## [Peer Review File · Nature Communications]

Reviewers' Comments:

Reviewer #1:

Remarks to the Author:

The current manuscript aims to investigate the dynamics of dosage compensation of the X chromosome during embryonic development. Using single cell expression analysis of hybrid embryos, the authors are able to assay how X upregulation and how X-inactivation occur during early stages of embryogenesis. They combine their expression studies with allele-resolved chromatin accessibility to define more fully investigate chromatin changes that result from dosage compensation. Their major findings include: (1) X-upregulation fine tunes X-linked dosage in a sex- and lineage-specific manner, (2) females with two active X's lack upregulation and express each allele at a basal level, and (3) X-upregulation is driven by transcriptional burst frequency and seems to be influenced by X-inactivation. Their conclusions are well supported by the data and provide a detailed analysis of the timing and dynamics of dosage compensation during imprinted X inactivation and random X inactivation.

Overall, the manuscript presents interesting findings that may lead to a revised model of X dosage compensation events.

There are a few points which should be addressed.

1. The authors culture ES cells in 2i +LIF media. There have been studies that show that those conditions can result in widespread DNA methylation loss (Werner et al *Biology of Sex Differences* 2017; Habibi et al *Cell Stem Cell* 2013; von Meyenn et al *Mol. Cell* 2016; Hackett et al *Stem Cell Reports* 2013; Ficz et al *Cell Stem Cell* 2013; Yagi et al *Nature* 2017). Aberrantly low DNA methylation of the autosomes in 2i conditions may lead to confounding results in X:A ratios in mESCs.

2. Are the iXCI studies done in cells from reciprocal crosses? The dynamics of XCI are slightly different between strains due to differences in Xce (Calaway et al *PLoS Genetics* 2013). In terms of imprinted XCI, could this be a confounding factor?

Minor concerns:

1. Is Xs defined in the manuscript or legends? Xs should be defined in the legend for Figure 2
2. Other concerns are more for aesthetics. Some figures are crowded and some figure keys are hard to see (for example Figure 4h).

Reviewer #2:

Remarks to the Author:

The manuscript by Lentini et al reported a thorough analysis of X-upregulation and X-inactivation through a development process in mouse by using allele-resolved single-cell RNA-seq combined with chromatin accessibility profiling. The data they created have changed the concept of dosage compensation in mammals as is taught in textbooks. For example, the two alleles that are not silenced in female both expressed in a basal level as autosomal genes show and the female inactive X genes exhibit upregulation of the single active X copy. Therefore, I recommend that *Nature Communication* publish a version of this work with some revision in presentation as soon as possible. The current version is very hard to read for its densely presented information.

1. The clustering of PCA analyses as shown in Figure 2B and 2F were based on a low proportion of the total variation, 2B: 28% (11% + 17%) total variance; 2F: 23% (7% + 16%) total variance. How can these clustering patterns be trustable while the vast majority portion of other variation components may suggest otherwise?

2. Too much data and results shown in Figures 1-3 are distracting and making it difficult to find the most important ones. 1/2 to 1/3 should be moved to supplementary data or supplementary

information. For example, Fig. 1A is not helpful for understanding how they worked out their analysis because it is understandably too concise when putting into Figure 1. If it is moved to supplementary information, more detailed information can be added to explain how the analyses were done.

3. The introduction does not do enough to explain why this research is new and important: the conventional picture from many in general biology and evolution is that mammals achieve dosage compensation by silencing one X in female and others like *Drosophila* by upregulation, hypertranscription, from the X in male. They should tell how this conventional notion was formed and why it is incorrect or just partially correct given the previous research systems.

Reviewer #3:

Remarks to the Author:

In this interesting paper Lentini and colleagues address the modalities of X upregulation during early mouse development when the two waves of X inactivation (imprinted and random) take place in mouse. The authors had previously shown that X upregulation is associated with increased frequency of transcription bursts. In the current analysis they show that prior to X inactivation when cells have two active X chromosomes, there is no apparent X upregulation. Yet, the combined expression of both active alleles is higher than that of the single upregulated active allele once X inactivation silences the other allele. Importantly, they demonstrate that preventing X inactivation prevents X upregulation. Thus, in females, X upregulation appears dependent on X inactivation, explaining the two waves of X upregulation.

The findings are novel and the experiments carefully done to demonstrate these changes during development of male and female embryos. This part of the manuscript is very strong. However, the authors go on to argue that the modulations of burst frequency are associated with changes in chromosomal compartments as shown by Hi-C. These experiments are utterly unconvincing. The differences we are meant to see on contact maps are not at all evident. It has been previously reported that the entire genome becomes more compartmentalized during early development. That the active X chromosome changes in a different manner from the autosomes is possible, but not demonstrated here. There is no quantitative analysis of TADs, loops, or A/B compartments done. This analysis should be the focus of a separate in depth study.

The authors proposed a model for the elastic changes in X upregulation depending on the X inactivation status, which is in part supported by their data. However, the models proposed include statements that are not based on experiments done here. For example, they proposed a transfer of transcription factors from the future inactive X to the single active X in females, with no supporting evidence.

Specific points

1. The figures are very busy and the legends should be clarified. References to figures in the text and the legends often leave the reader wondering how to interpret the figures. For example, (one of many) what are we to conclude about Figure 3b?

2. The authors should define Xs in the text and in the figures and their legends.

3. Line 236-238: The authors argue that the lower allelic expression in XaXa is unlikely due to repression based on findings of accessible chromatin. The argument is not very convincing, as there are examples of genes with positive chromatin accessibility, which are not expressed.

4. Line 295-297: increased transcription factor concentration on the hyperactive Xa has not been demonstrated anywhere in this paper. Many other mechanisms could increase expression.

5. Figure 5a at the top nicely shows changes in X upregulation during development. However, on the bottom the authors show a loop regulating bursts of transcription, which has not been convincingly shown in the manuscript.

6. Figure 5b shows possible models for X upregulation, which is fine. However, in the legend the authors claim that the concentration of transcription factors is shifted to the Xa from the Xi. However, this has not been demonstrated in the paper. Thus, there is again over-interpretation of the data at hand.

REVIEWER COMMENTS

Reviewer #1 (Remarks to the Author):

The current manuscript aims to investigate the dynamics of dosage compensation of the X chromosome during embryonic development. Using single cell expression analysis of hybrid embryos, the authors are able to assay how X upregulation and how X-inactivation occur during early stages of embryogenesis. They combine their expression studies with allele-resolved chromatin accessibility to define more fully investigate chromatin changes that result from dosage compensation. Their major findings include: (1) X-upregulation fine tunes X-linked dosage in a sex- and lineage-specific manner, (2) females with two active X's lack upregulation and express each allele at a basal level, and (3) X-upregulation is driven by transcriptional burst frequency and seems to be influenced by X-inactivation. Their conclusions are well supported by the data and provide a detailed analysis of the timing and dynamics of dosage compensation during imprinted X inactivation and random X inactivation.

Overall, the manuscript presents interesting findings that may lead to a revised model of X dosage compensation events.

There are a few points which should be addressed.

We thank the reviewer for the comments and the positive remarks. We have addressed all points raised. The changes in the manuscript text and figure legends are highlighted in yellow color to facilitate the review (some of the changes in figure legends are not highlighted as we restructured and removed several of the panels, based on the reviewers' comments). We hope that you now find our study ready for publication.

1. The authors culture ES cells in 2i +LIF media. There have been studies that show that those conditions can result in widespread DNA methylation loss (Werner et al *Biology of Sex Differences* 2017; Habibi et al *Cell Stem Cell* 2013; von Meyenn et al *Mol. Cell* 2016; Hackett et al *Stem Cell Reports* 2013; Ficiz et al *Cell Stem Cell* 2013; Yagi et al *Nature* 2017). Aberrantly low DNA methylation of the autosomes in 2i conditions may lead to confounding results in X:A ratios in mESCs.

We agree with the reviewer that X:A ratios are difficult to interpret – and that ratios could potentially be affected by widespread changes unrelated to XCU. Fortunately, in this study, we moved beyond the indirect X:A measure in the key parts of our analyses and measured X-upregulation directly as the single-allele X-chromosome expression changes as XCU establishes on the X-chromosome. Further, we already investigated the potential effect of 2i+Lif media: For cells grown in 2i+LIF or Activin+FGF conditions we show data stratified by growth condition in **Fig. 3e** and **Supplementary Fig. 1g** where our findings on XCU are consistent between the two – *i.e.* XCU of same/similar magnitude is observed in cells having one active X-allele and absent in cells with two active X-chromosomes. The observations made in our *in vitro* system was further recapitulated in the *in vivo* data (which does not rely on 2i+LIF conditions) as well as in our re-analysis of published 2i/Lif-withdrawal data (*e.g.* **Supplementary Fig. 1h-j and 3c**).

2. Are the iXCI studies done in cells from reciprocal crosses? The dynamics of XCI are slightly different between strains due to differences in Xce (Calaway et al *PLoS Genetics* 2013). In terms of imprinted XCI, could this be a confounding factor?

Our blastocyst data (subject to iXCI) are from reciprocal crosses (represented at roughly 50/50 ratio for females) where both crosses showed consistent X-upregulation on the active allele in parallel with iXCI on the second allele independently of X-allele origin (e.g. see **Fig. 2b**). The Xist knockout (paternal deletion) data was derived from a single cross, but both the wildtype and knockout cells behaved similarly to the reciprocal blastocyst data in terms of XCU. For random XCI, our data shows the expected skew in cells frequencies with C57 allele becoming inactive (Calaway PLOS Genetics 2013 as well as Chadwick Genetics 2006, Thorvaldsen Genetics 2012 & Reinius Nature Genetics 2016), as reflected in the number of Xa(cast)Xi(c57) vs. Xi(cast)Xa(c57) cells (see **Fig. 1h**), but regardless of strain-of-origin the alleles shows the same magnitude and dynamics of XCU (see for example main Figure 1i and 3f-g). The exact timing of imprinted XCI on C57 and CAST alleles has been characterized before (Borensztein 2017 NSMB; e.g. **Borensztein Fig. 1d and 2**) and found to be similar. Moreover, slight timing between strains would not affect our conclusion that XCU occurs upon XCI (imprinted as well as random) in females.

We have also included the following text in the methods to clarify the XCI skew in our data, citing the above references:

“Due to differences in X-controlling elements (Xce) between C57 and CAST strains, the C57 allele is preferentially inactivated during random XCI [References] which is observable in our data as the number of Xa_{C57}Xi_{CAST} vs. Xi_{C57}Xa_{CAST} cells.”

Minor concerns:

1. Is Xs defined in the manuscript or legends? Xs should be defined in the legend for Figure 2

Thank you for noticing the lack of “Xs” definition in figures. We have now added this information.

2. Other concerns are more for aesthetics. Some figures are crowded and some figure keys are hard to see (for example Figure 4h).

We appreciate the input on the aesthetics as we want to maximize the readability of our study. We have therefore moved several panels into Supplementary figures as well as resized and spaced out the panels in the updated main figures. In some figures we have removed unnecessary information and recolored plots to be more coherent. The larger changes to panels are the following:

Figure1 - Simplified, increased size and improved colors in panels **a,b,c** and **i**. Replaced schematic figure in panel **j** with actual data, including error bars.

Figure2 - Removed panels **a,b,f** and improved the annotations for the new panel **a**.

Figure3 - Moved panels **b,e** and part of **d** to supplementary, split density and escapee plots into new panels **b** and **c**.

Figure4 - Moved panel **i** to supplementary and increased the size of panel **h** to better see details.

Figure5 - Removed the schematic DNA loop in panel **c** and replaced it with a dosage schematic.

We decided on keeping schematic figures **Fig. 1a** and **Fig. 4a** as we believe they provide an overview of the study design of the new experiments conducted in this study where in-depth details are given in the methods but have also included new **Supplementary Data 1** that contain annotated details of each cell included in these experiments.

Reviewer #2 (Remarks to the Author):

The manuscript by Lentini et al reported a thorough analysis of X-upregulation and X-inactivation through a development process in mouse by using allele-resolved single-cell RNA-seq combined with

chromatin accessibility profiling. The data they created have changed the concept of dosage compensation in mammals as is taught in textbooks. For example, the two alleles that are not silenced in female both expressed in a basal level as autosomal genes show and the female inactive X genes exhibit upregulation of the single active X copy.

Therefore, I recommend that Nature Communication publish a version of this work with some revision in presentation as soon as possible. The current version is very hard to read for its densely presented information.

We thank the reviewer for the constructive comments and remarks, and we have addressed the questions raised point-by-point below. The changes in the manuscript text and figure legends are highlighted in yellow color to facilitate the review (some of the changes in figure legends are not highlighted as we restructured and removed several of the panels, based on the reviewers' comments). We hope that you agree that all concerns have been fully responded to and that our study is ready for publication.

1. The clustering of PCA analyses as shown in Figure 2B and 2F were based on a low proportion of the total variation, 2B: 28% (11% + 17%) total variance; 2F: 23% (7% + 16%) total variance. How can these clustering patterns be trustable while the vast majority portion of other variation components may suggest otherwise?

No clustering was performed for these cells as the cell identity was already known, either from manual dissection and picking or previous in-depth categorization (see our previous work in Deng et al. 2014 Science and Cheng et al. 2019 Cell reports for details). As the visualization in old Fig. 2b is not important for our results we have removed it to increase figure clarity as suggested by the Reviewers, including the next point raised by the Reviewer.

2. Too much data and results shown in Figures 1-3 are distracting and making it difficult to find the most important ones. 1/2 to 1/3 should be moved to supplementary data or supplementary information. For example, Fig. 1A is not helpful for understanding how they worked out their analysis because it is understandably too concise when putting into Figure 1. If it is moved to supplementary information, more detailed information can be added to explain how the analyses were done.

We appreciate the input on the aesthetics as we want to maximize the readability of our study. We have therefore moved several panels into Supplementary figures as well as resized and spaced out the panels in the updated main figures. In some figures we have removed unnecessary information and recolored plots to be more coherent. The larger changes to panels are the following:

Figure1 - Simplified, increased size and improved colors in panels **a,b,c** and **i**. Replaced schematic figure in panel **j** with actual data, including error bars.

Figure2 - Removed panels **a,b,f** and improved the annotations for the new panel **a**.

Figure3 - Moved panels **b,e** and part of **d** to supplementary, split density and escapee plots into new panels **b** and **c**.

Figure4 - Moved panel **i** to supplementary and increased the size of panel **h** to better see details.

Figure5 - Removed the schematic DNA loop in panel **c** and replaced it with a dosage schematic.

We decided on keeping schematic figures **Fig. 1a** and **Fig. 4a** as we believe they provide an overview of the study design of the new experiments conducted in this study where in-depth details are given

in the methods but have also included new **Supplementary Data 1** that contain annotated details of each cell included in these experiments.

3. The introduction does not do enough to explain why this research is new and important: the conventional picture from many in general biology and evolution is that mammals achieve dosage compensation by silencing one X in female and others like *Drosophila* by upregulation, hyper-transcription, from the X in male. They should tell how this conventional notion was formed and why it is incorrect or just partially correct given the previous research systems.

Thanks for highlighting the need to phrase this more directly. We now state the conventional model explicitly in the introduction:

“This stands as the prevailing evolutionary hypothesis of mammalian dosage compensation²⁻⁴. In addition, XCU followed by XCI is also believed to be the developmental sequence of events. Specifically, the conventional model of mammalian dosage compensation assumes an initial state of biallelically upregulated X-chromosomes in female embryonic cells upon which XCI subsequently corrects the expression dosage by silencing one X allele. However, despite being central for the mechanistic understanding of how cells achieve dosage compensation, the developmental dynamics of XCU have so far not been thoroughly characterized in a mammal.”

Based on the last part of the reviewer comment, we have also added a sentence in Abstract that clearly highlights this:

“Importantly, this conflicts the conventional dosage compensation model in which naïve female cells are subject to biallelic X-upregulation followed by X-inactivation.”

We however think that bringing up the models of evolutionarily unrelated dosage compensation systems in more distant species such as *Drosophila* makes the introduction too long and complex, and we opted to not go into this in the Introduction.

Reviewer #3 (Remarks to the Author):

In this interesting paper Lentini and colleagues address the modalities of X upregulation during early mouse development when the two waves of X inactivation (imprinted and random) take place in mouse. The authors had previously shown that X upregulation is associated with increased frequency of transcription bursts. In the current analysis they show that prior to X inactivation when cells have two active X chromosomes, there is no apparent X upregulation. Yet, the combined expression of both active alleles is higher than that of the single upregulated active allele once X inactivation silences the other allele. Importantly, they demonstrate that preventing X inactivation prevents X upregulation. Thus, in females, X upregulation appears dependent on X inactivation, explaining the two waves of X upregulation.

The findings are novel and the experiments carefully done to demonstrate these changes during development of male and female embryos. This part of the manuscript is very strong. However, the authors go on to argue that the modulations of burst frequency are associated with changes in chromosomal compartments as shown by Hi-C. These experiments are utterly unconvincing. The differences we are meant to see on contact maps are not at all evident. It has been previously reported that the entire genome becomes more compartmentalized during early development. That the active X chromosome changes in a different manner from the autosomes is possible, but not demonstrated here. There is no quantitative analysis of TADs, loops, or A/B compartments done. This analysis should be the focus of a separate in depth study.

The authors proposed a model for the elastic changes in X upregulation depending on the X inactivation status, which is in part supported by their data. However, the models proposed include statements that are not based on experiments done here. For example, they proposed a transfer of transcription factors from the future inactive X to the single active X in females, with no supporting evidence.

We thank the reviewer for the constructive and useful comments. We agree with the reviewer's remarks that our results on the dynamics of X-upregulation (key findings) are both novel and solid. We also agree that the part of the manuscript where we probe into potential epigenetic modulators of increased bursting is more explorative, and that in-depth characterization of DNA-DNA interactions upon XCU deserves a dedicated study of its own. We nonetheless think that the current Hi-C data, albeit lacking the resolution to investigate specific elements at the allelic resolution, are valuable as a starting point. Thus, we have toned down the results and conclusions on the Hi-C data. We furthermore moved the Hi-C panel (previously in **Fig. 4**) into the **Supplementary information**. Regarding the section in the **Discussion** about transcription factors potentially shifting from one allele to the other, we explicitly clarified that this is a hypothesis and not to be interpreted as a result.

The changes in the manuscript text and figure legends are highlighted in yellow color to facilitate the review (some of the changes in figure legends are not highlighted as we restructured and removed several of the panels, based on the reviewers' comments). We hope that you now find our study ready for publication.

Specific points

1. The figures are very busy and the legends should be clarified. References to figures in the text and the legends often leave the reader wondering how to interpret the figures. For example, (one of many) what are we to conclude about Figure3b?

We appreciate the input on the aesthetics as we want to maximize the readability of our study. We have therefore moved several panels into Supplementary figures as well as resized and spaced out the panels in the updated main figures. In some figures we have removed unnecessary information and recolored plots to be more coherent. The larger changes to panels are the following:

Figure1 - Simplified, increased size and improved colors in panels **a,b,c** and **i**. Replaced schematic figure in panel **j** with actual data, including error bars.

Figure2 - Removed panels **a,b,f** and improved the annotations for the new panel **a**.

Figure3 - Moved panels **b,e** and part of **d** to supplementary, split density and escapee plots into new panels **b** and **c**.

Figure4 - Moved panel **i** to supplementary and increased the size of panel **h** to better see details.

Figure5 - Removed the schematic DNA loop in panel **c** and replaced it with a dosage schematic.

We decided on keeping schematic figures **Fig. 1a** and **Fig. 4a** as we believe they provide an overview of the study design of the new experiments conducted in this study where in-depth details are given in the methods but have also included new **Supplementary Data 1** that contain annotated details of each cell included in these experiments.

2. The authors should define Xs in the text and in the figures and their legends.

Thank you for noticing the lack of “Xs” definition in figures. We have now added this information.

3. Line 236-238: The authors argue that the lower allelic expression in XaXa is unlikely due to repression based on findings of accessible chromatin. The argument is not very convincing, as there are examples of genes with positive chromatin accessibility, which are not expressed.

We agree that we have not exhaustively ruled out all possibilities related to partial repression of the XaXa state and have therefore removed this statement.

4. Line 295-297: increased transcription factor concentration on the hyperactive Xa has not been demonstrated anywhere in this paper. Many other mechanisms could increase expression.

We realized that this sentence could be mistaken as a results statement in combination with the legends for **Fig. 5b** (as pointed out in Reviewer comment 6) and have clarified these sentences as mere hypotheses. We also removed the part of the **Fig. 5** legend referring to this. We still however see value in mentioning some of the potential mechanisms in the **Discussion** that could be pursued in future studies.

5. Figure 5a at the top nicely shows changes in X upregulation during development. However, on the bottom the authors show a loop regulating bursts of transcription, which has not been convincingly shown in the manuscript.

We thank the reviewer for the remarks on **Fig. 5a**. We agree that the DNA-loop illustration (old panel **5c**) was somewhat speculative as well as unnecessary to include. We therefore removed this illustration and replaced it with a schematic clearly illustrating that XaXa dosage > Xa”Xi, which is a key finding in our study and important to highlight in the model figure as it explains previous measurements of increased X-chromosome expression in ECS yet lack of XCU in such cells.

6. Figure 5b shows possible models for X upregulation, which is fine. However, in the legend the authors claim that the concentration of transcription factors is shifted to the Xa from the Xi. However, this has not been demonstrated in the paper. Thus, there is again over-interpretation of the data at hand.

We have removed this speculation from the legend, see answer to point 4.

We express our gratitude to the three constructive reviewers. We believe that we have addressed all concerns raised fully and hope that you now find our study ready for publication.

Reviewers' Comments:

Reviewer #1:

Remarks to the Author:

The author's have adequately addressed my concerns.

Reviewer #2:

Remarks to the Author:

I have read the revision and responses to my previous review. I am satisfactory with the revision, which I think has addressed my concerns well. The ms reads more smooth than the first version. I am happy to recommend this version to NC for publication.

Reviewer #3:

Remarks to the Author:

Reinius and colleagues have responded to most of the comments of the reviewers. They also clarified complex figures, making the paper more readable.

REVIEWERS' COMMENTS

Reviewer #1 (Remarks to the Author):

The author's have adequately addressed my concerns.

We thank Reviewer #1 for taking their time to review and provide constructive criticism to our manuscript which we feel have greatly improved the final product.

Reviewer #2 (Remarks to the Author):

I have read the revision and responses to my previous review. I am satisfactory with the revision, which I think has addressed my concerns well. The ms reads more smooth than the first version. I am happy to recommend this version to NC for publication.

We thank Reviewer #2 for taking their time to review and provide constructive criticism to our manuscript which we feel have greatly improved the final product.

Reviewer #3 (Remarks to the Author):

Reinius and colleagues have responded to most of the comments of the reviewers. They also clarified complex figures, making the paper more readable.

We thank Reviewer #3 for taking their time to review and provide constructive criticism to our manuscript which we feel have greatly improved the final product.